# A Novel Adaptive FCM with Cooperative Multi-Population Differential Evolution Optimization

**Amit Banerjee \* and Issam Abu-Mahfouz**

Mechanical Engineering, School of Science Engineering and Technology, Pennsylvania State University Harrisburg, Middletown, PA 17057, USA
\* Correspondence: aub25@psu.edu

**Abstract:** Fuzzy c-means (FCM), the fuzzy variant of the popular *k*-means, has been used for data clustering when cluster boundaries are not well defined. The choice of initial cluster prototypes (or the initialization of cluster memberships), and the fact that the number of clusters needs to be defined *a priori* are two major factors that can affect the performance of FCM. In this paper, we review algorithms and methods used to overcome these two specific drawbacks. We propose a new cooperative multi-population differential evolution method with elitism to identify near-optimal initial cluster prototypes and also determine the most optimal number of clusters in the data. The differential evolution populations use a smaller subset of the dataset, one that captures the same structure of the dataset. We compare the proposed methodology to newer methods proposed in the literature, with simulations performed on standard benchmark data from the UCI machine learning repository. Finally, we present a case study for clustering time-series patterns from sensor data related to real-time machine health monitoring using the proposed method. Simulation results are promising and show that the proposed methodology can be effective in clustering a wide range of datasets.

**Keywords:** fuzzy c-means; initial cluster prototypes; optimal number of clusters; differential evolution; sparse sampling; Hopkins statistic; cluster validity indices





## 1. Introduction

Clustering is an unsupervised learning method that seeks to partition objects in a dataset into several natural groupings called clusters such that objects within a cluster tend to have similar attributes while objects belonging to different clusters have dissimilar attributes. In other words, clustering methods that can produce self-similar groups characterized by intra-group homogeneity and heterogeneity across groups, tend to create good natural partitions of the dataset. This is conceptually different from supervised classification or discriminant analysis where labeled data is used to train a classifier that is eventually used to classify unlabeled data. Clustering is useful in several exploratory data analyses, visualization, decision making, and machine-learning applications such as data mining, image segmentation, and pattern recognition. There are several ways to categorize different clustering methodologies, one of the most common is categorizing clustering methods as hierarchical clustering or partitional clustering methods [1]. Hierarchical clustering methods organize data into hierarchical structures of partitions starting from singleton clusters (each object in the data is its own cluster) to one cluster over the entire data and every structure in between. Partitional clustering methods, on the other hand, produce a single partition of data for a pre-specified number of clusters based on the optimization of a pre-determined clustering criterion. While defining a suitable proximity measure (or a distance measure) is all that is often required for hierarchical clustering, partitional clustering methodologies require specifying clustering criteria (objective functional), number of clusters sought, and other parameters depending on the type of algorithm being used.

We focus on fuzzy clustering in this paper. Unlike hard clustering where an object belongs to a single cluster, the notion of graded belongingness of objects to all clusters

is key to fuzzy clustering [2]. The most widely used fuzzy clustering algorithm is fuzzy c-means [3,4] which is the fuzzy equivalent of the hard *k*-means algorithm. It uses a membership function grade to associate each object in the data to clusters. The clusters themselves are defined as cluster centers or cluster prototypes or simply centroids (a centroid is an object instance which is most central to the cluster being described). Objects are assigned to the clusters based on their membership in the cluster. The alternating optimization (AO) algorithm is the most popular implementation scheme for FCM [5]. The prototypes are initialized, either randomly or procedurally. At each optimization step, the partition memberships and the prototypes are updated, until a pre-defined stopping criterion is met such as when prototypes have stabilized. The number of clusters to be found is defined *a priori*. However, FCM is sensitive to the initialization of prototypes and the fact that in many applications it is difficult to *a priori* define the number of clusters to be found.

A novel method to find initial cluster prototypes (initialization) while concurrently determining the optimal number of clusters to be found, is presented in this paper. Contributions of this paper are as follows.

(1) A subset method is introduced to create the best subset of the dataset which still preserves the underlying structure of the original dataset. This is performed using a measure based on sparse sampling of the data, a concept widely used in clusterability studies of datasets. The reduced subset is then used instead of the original dataset to initialize cluster prototypes and find the optimal number of clusters.

(2) A coevolutionary scheme is presented which evolves candidate solutions that encode for both cluster prototypes as *d*-dimensioned real-valued numbers as well as the number of clusters. A multi-population differential evolution algorithm is presented with each population using a randomly assigned variant of the differential evolution algorithm. An elite population that does not evolve candidate solutions directly but participates in the evolution of the other populations is fundamental to the proposed scheme.

(3) A cluster-validity index is used as the fitness function to guide the evolution. The cluster validity index is calculated by first computing memberships using the information encoded in the evolutionary candidate vector and then using the memberships to find the index as a single-step operation. Using the reduced subset method, the implementation is less resource intensive compared to using the entire dataset.

The rest of the paper is organized as follows. We start with providing a background of FCM with a review of recent literature in the determination of the optimal number of clusters and the methods for cluster prototype initialization in Section 2. In Section 3 we present the subset selection method, which is central to the proposed method, followed by the multi-population coevolutionary differential evolution framework in Section 4. In Section 5, we present simulation results using synthetic data and datasets from the University of California Irvine's Machine Learning (UCI ML) repository. In Section 6, we present a case study with a real-world dataset from the domain of machine health monitoring. In Section 7, we conclude the paper and present some directions for future research.

## 2. Background

Fuzzy c-means or FCM is a partitional clustering algorithm based on the notion of fuzzy membership of objects in a dataset to a cluster. FCM is useful when the natural partitions in a dataset are not evident or very well defined. A data object $x_k$ has a certain membership $u_{ik}$ (which takes values between zero and one, not including 0 or 1) in a cluster $C_i$, which is seen as the partial (fuzzy) belongingness of the data point to that cluster, subject to the constraint that the sum of memberships across all clusters is unity and the

contribution of memberships of all data points to any particular cluster is always less than the cardinality of the dataset $n$ ($k = 1, 2, \ldots, n$).

$$\sum_{i=1}^{c} u_{ik} = 1; \ 0 < \sum_{k=1}^{n} u_{ik} < n \tag{1}$$

The fuzzy sum-of-squared-error objective function is the least squares estimator function,

$$J_{FCM} = \sum_{i=1}^{c} \sum_{k=1}^{n} u_{ik}^{m} ||x_k - v_i||^2 \tag{2}$$

The exponent $m$ is called the fuzzifier which determines the fuzziness of the partition, $m \in [1, \infty)$ and $||x_k - v_i||_2 = (x_k - v_i)'A(x_k - v_i)$ is the distance between cluster prototype $v_i$ and data object $x_k$ where $A$ is the norm matrix. The identity norm matrix $A = I$ yields Euclidean distance and results in spherical clusters while other norms are used for elliptical clusters, etc. In this paper, the Euclidean distance has been used. The FCM-AO minimizes the functional in Equation (2) by iteratively calculating cluster prototypes and updating memberships until there is no change in the termination criterion, usually as measured by a change in memberships between two successive iterations.

$$v_i = \frac{\sum_{k=1}^{n} u_{ik}^{m} x_k}{\sum_{k=1}^{n} u_{ik}^{m}} \quad \forall i = 1, 2, \ldots, c \tag{3}$$

$$u_{ik} = \frac{1}{\sum_{j=1}^{c} \left[ \frac{||x_k - v_i||^2}{||x_k - v_j||^2} \right]^{\frac{1}{m-1}}} \tag{4}$$

The equations presented for FCM are from [4]. The algorithm often starts by initializing cluster prototypes randomly, then calculating memberships using Equation (4), followed by recalculating prototypes using Equation (3) until convergence. The final membership matrix $\mathbf{U} = [u_{ik}]_{c \times n}$ is sensitive to the initialization of cluster prototypes. In addition, the iterative optimization scheme assumes that the number of clusters $c$ is known. The only parameter to tune is the fuzzifier $m$ which depending on the fuzzification required is often set to $m = 2$ for most clustering problems (especially one involving spherical clusters).

### 2.1. Determination of Optimal Number of Clusters

For two-dimensional datasets, the number of clusters can be determined by simple visualization. For higher dimensional data, the dataset can be mapped to a two-dimensional plane using dimensionality reduction techniques such as principal component analysis (PCA), kernel, locally linear embedding (LLE), diffusion maps, and sparse dictionary representations. These techniques do not, however, provide a mapping that conserves the internal partitional structure of the dataset. More popular are cluster validity methods that quantify the goodness of a partition after the clustering method produces the partition. Different partitions for different values of $c$ are created and then the goodness of partition is compared using a measurable index. The value of the index reaching a maximum or minimum or an inflection point is often a good indicator of the goodness of partition and the corresponding number of clusters is the optimal number of clusters. We use several of these cluster validity measures in this paper. However, these are used not to determine an optimal value for $c$, but instead to compare the performance of different algorithms.

The first cluster validity index was proposed by Zadeh called the degree of separation which was later refined as the concept of partition coefficient and the closely related partition entropy by Bezdek [6]. Lee proposed a fuzzy clustering validity index using the distinguishableness of clusters measured by the object proximities [7]. Based on Shannon

entropy and fuzzy variation theory, Zhang and Jiang proposed a fuzzy clustering validity index taking into account of the geometry structure of the dataset [8]. Saha et al. presented an algorithm based on differential evolution for automatic cluster detection, which evaluated the validity of the clustering result [9]. Yue et al. partitioned the original data space into a grid-based structure and proposed a cluster separation measure based on grid distances [10]. Based on the idea that a natural partition is one that creates well-separated compact clusters, measures of compactness and separation were proposed as part of several cluster validity measures. These include Fukuyama–Sugeno index *FS* [11], Xie–Beni index *XB* [12], Bensaid index *SC* [13], Tang index $V_T$ [14], Kwon index $V_K$ [15], *PBMF* index [16], *PCAES* index [17], *WL* index [18], Wang index $V_W$ [19], *CWB* index [20], and Zhu index $V_z$ [21].

Since these measures compare and quantify the goodness of the partition after the partition is generated, they are not practical for problems involving large datasets (high cardinality *n*) or high number of dimensions *d*. Lately, some new clustering algorithms have been proposed that adjust the number of clusters while the partition is being generated. Competitive agglomeration [22] and split hierarchical clustering [23] have been used to guide the optimization of the number of clusters in a partitional clustering process. A similarity-based clustering method that combines single-point iteration with hierarchical clustering to determine the number of clusters is proposed in [24]. A Mercer kernel-based clustering [25] estimated the number of clusters by the eigenvectors of a kernel matrix. A clustering algorithm based on maximal $\theta$-distant subtrees [26] detected any number of well-separated clusters of any shape.

### 2.2. Initialization of Cluster Prototypes

In most implementations of FCM-AO, the cluster centers are randomly initialized. The random initialization affects the accuracy and running time of the algorithm, especially in datasets where natural clusters overlap. Based on the initial choice of cluster prototype, the algorithm can either quickly converge or get trapped in local minima. A subtractive clustering algorithm has been used to find the initial cluster centers in [27,28]. However, there are several parameters that need to be set to get the desired outcome, which are not trivial. A cluster prototype initialization based on a density cluster algorithm is presented in [29]. A fuzzy entropy algorithm is proposed with a hybrid FCM algorithm to initialize cluster centers in [30]. An improved FCM algorithm based on morphological reconstruction and affiliation filtering (FRFCM) which can determine the number of clusters while optimizing initial cluster prototypes has been presented in [31].

Metaheuristic methods such as evolutionary algorithms including genetic algorithms [32,33], particle swarm optimization, and differential evolution [34] have been used for cluster prototype initialization. A detailed recent review of metaheuristics used to solve the problem of cluster prototype initialization for FCM in the area of image segmentation can be found in [35]. A single population differential evolution algorithm called automatic differential evolution-based fuzzy clustering (ADEFC) was used to optimize initial cluster prototypes and number of clusters in [9]. The multiple population genetic algorithm (MPGA) to optimize cluster prototypes using multiple evolving populations was proposed in [36]. Each subpopulation is allowed to evolve independently; however, a migration operator is used to relate the individually evolving populations. Recently, the derivative multi-population genetic algorithm (DMGA) was proposed in [37] which first initializes the population by using a derivative operator before each subpopulation is evolved using canonical genetic operators. The probabilities of the genetic operators are dynamically selected by an adaptive probability fuzzy control operator. The quality of the initial cluster prototypes is shown to be superior to those found using MPGA. However, these methods are very resource intensive for big datasets.

## 3. Subset Selection

A co-evolutionary algorithm recognizes the diversity of the possible candidate solutions in the population. It emphasizes the correlation between the domain of the problem being solved and the candidate solutions. Evolutionary algorithms are prone to get trapped in local optima, and performance depends on fine tuning a few or many parameters. Co-evolutionary systems, on the other hand, have been shown to have the ability to avoid local optima by dividing the solution space and also use different evolutionary strategies to support either a competitive system or a cooperative system. Their strength lies in the divide-and-conquer decomposition strategy and in their implicit parallelism. These algorithms can be divided into competitive co-evolutionary systems and cooperative co-evolutionary systems and include systems such as co-evolution of predator and prey systems, competing species, among others [38–40]. These systems can balance the exploration and exploitation capability by utilizing cooperative or competitive mechanisms among different subpopulations using different strategies for different subpopulations. They can also improve convergence properties (not necessarily speed) by facilitating information exchange among the subpopulations.

There are two important reasons for using coevolutionary algorithms for clustering problems: (1) evolution of candidate solutions on a subset of the larger dataset improves the convergence properties of the individual subpopulation. This is because a judiciously selected subset of a larger dataset can capture the structure of the larger dataset and since it is smaller than the original dataset, an evolutionary algorithm can converge faster. (2) Coevolution, both competitive and cooperative, ensures that excellent candidates which in a single-strategy one-population evolution may not be able to evolve for several generations which is an implicit strain on resources, now have a better chance of getting selected for evolution.

The proposed co-evolutionary framework is described here. The dataset of size $n$ to be clustered is first divided into SP distinct subsets such that in each subset there are $n/$SP objects. If $n$ is not evenly divisible by SP, then the largest number $< n$ evenly divisible by SP is used. For example, if $n = 528$ and SP = 10, we select 520 objects at random from the dataset and divide them into 10 subsets with 52 objects in each. Every subset is mutually exclusive so that a data object is available to only one subset. This ensures an even coverage of the entire dataset. The algorithm to create mutually exclusive subsets of the data is shown as Algorithm 1.

---

**Algorithm 1**: Dividing dataset into subsets

---

**Input:** Dataset **data** of size n and dimensions d: X = {$x_1$, $x_2$, ... $x_n$}, where $x_j$ = {$x_{j1}$, $x_{j2}$, ... $x_{jd}$}, SP

1. Calculate rem = remainder(n/SP). If rem ! = 0, discard rem datapoints randomly from **data** and retain (n − rem) datapoints as **data**
2. **For** k = 1 to SP
3. Select (n − rem)/SP datapoints at random from **data** to populate subset $P_k$
4. Remove the same (n − rem)/SP points from data: **data** = **data** − $P_k$
5. **End**

**Output:** SP Subsets P = {$P_1$, $P_2$, ... $P_{SP}$}, rem

---

Of these distinct datasets, only the one with a similar underlying structure as the original dataset is then selected. This is performed by employing a sparse sampling statistic based on a test of spatial randomness. Sparse sampling tests are based on sampling origins randomly assigned in a sampling window and the underlying structure of the data can be quantified by nearest neighbor measurement of sampling origins to points and comparing them to the random nearest neighbor measurement of paired points within the sampling window. Several tests involving sampling origins have been proposed in the literature, mostly from the field of biological statistics, based on a multitude of tests such as the Hopkins [41,42], Holgate [43], T-square [44], Eberhardt [45] and Cox-Lewis [46]. These

have been used as a test of clusterability, clustering tendency, and cluster validation [47,48]. In this paper, we use the Hopkins statistic which is by far the easiest of the statistical measures to implement. For a dataset $X$ of size $n$ in $d$-dimensions, $n_0$ sampling origins $Y$ are placed at random in a sampling window such that $n_0 << n$. The sampling origins $Y$ are $d$ dimensional as well. Two types of distances are defined: $u_k$ is the distance of the sampling origin $y_k$ to its nearest datapoint in $X$, and $w_k$ is the distance of a randomly chosen datapoint $x_k$ to its nearest neighbor in $X$. The Hopkins statistic [41] is defined as,

$$H = \frac{\sum\limits_{k=1}^{n_0} u_k^d}{\sum\limits_{k=1}^{n_0} u_k^d + \sum\limits_{k=1}^{n_0} w_k^d} \tag{5}$$

The statistic, therefore, compares the nearest-neighbor distribution of randomly selected locations in the dataset, to the nearest-neighbor distribution among objects in the data. If the underlying structure of dataset $X$ is random, then on an average the sum of $u_k$ would be equal to the sum of $w_k$ over $d$ dimensions and therefore $H$ will be close to 0.5. However, if dataset $X$ has distinct separable clusters (without any notion of how many clusters there are), then on an average the sum of $u_k$ will greatly exceed the sum of $w_k$ and therefore $H$ will be very close to unity. In most cases, real-world datasets will fall somewhere within this spectrum of complete underlying randomness and distinct separable clusters—as a result, Hopkins statistic $H$ will range between 0.5 and 1.0 for most datasets. If it is close to 0.5, the dataset is more random than clustered, and if close to 1.0 then there are distinct separable clusters. It is easy to see why the Hopkins statistic has been so popular for use as a measure of clustering tendency (the question of whether a dataset has an underlying cluster structure). In this paper, we use the statistic simply as a quantification of the underlying structure and compare it to the underlying structure of the subsets. The subset that under repeated spare sampling has a Hopkins statistic value close to that of the entire dataset most likely captures the underlying structure of the dataset. This is shown as Algorithm 2 below.

---

**Algorithm 2**: Identifying subset with similar underlying structure to the original dataset

---

**Input:** Dataset **data** of size $(n - rem)$, SP subsets $P = \{P_1, P_2, \ldots P_{SP}\}$ each of size $(n - rem)/SP$, number of sampling origins $n_0$ for **data**, convergence threshold $\varepsilon$

1. Populate $n_0$ datapoints at random in **data**
2. Calculate Hopkins statistic H for **data**
3. diff = $\varepsilon$, best_subset = $\varnothing$
4. **Repeat**
5. **For** k = 1 to SP
6. Populate $n_0/SP$ datapoints at random in $P_k$
7. Calculate Hopkins statistic $H_k$ for subset $P_k$
8. If $|H - H_k| < diff$, diff = $|H - H_k|$ and best_subset = k
9. **End**
10. **Until** (best_subset != $\varnothing$)

**Output:** Subset $P_{best\_subset}$

---

## 4. Proposed Co-Evolutionary Framework

A co-evolutionary framework is proposed to optimize the initial cluster prototypes and the number of clusters chosen. The most similar subset identified in Section 3 is then acted upon by subpopulations each running a different variant of the differential evolution (DE) algorithm. Candidate vectors are randomly created using a real-numbered vector representation and a masker scheme and then evolved using a DE based on a fitness criterion. In this section, we first describe the different variants of DE algorithms used in

this work, followed by the vector representation and masker scheme, and conclude with a discussion on the fitness function.

### 4.1. Differential Evolution Algorithms

Differential Evolution (DE) is a very popular population-based metaheuristic optimization technique used for multidimensional real-valued functions [49]. A distinct advantage of population-based methods is that they do not require gradient information for optimization as in the function being optimized (minimized or maximized) does not have to be differentiable. The technique uses a simple differential operator to create new candidate solutions by employing a one-to-one greedy competition between individuals in the population to move the population to optimal regions of the solution space. However, there is no guarantee that a global optimum will be attained. The differential operator is a combination of mutation and crossover operators used commonly in evolutionary algorithms to evolve populations. It uses the differences between randomly selected individuals in the population as the source of random variations for a third individual referred to as the target vector. The mutation operator is first used to generate a mutant vector by adding weighted difference vectors to the target vector. By computing the differences between two individuals from the population, the algorithm estimates the gradient in that zone rather than in a single point in the search space. There are many variations in the way the mutant vector is created and those used in this paper are described below. For a more detailed review of recent advances in differential evolution, the reader is referred to [50,51].

1.  DE/rand/1: The mutant vector for the new generation is generated by adding the weighted difference of two candidates (or vectors) in the present generation $g$ to the third vector known as the target vector in the present population.

$$\mathbf{m}_i(g+1) = \mathbf{p}_i(g) + F_s[\mathbf{p}_1(g) - \mathbf{p}_2(g)] \qquad (6)$$

where $\mathbf{p}_1$ and $\mathbf{p}_2$ are randomly chosen vectors from the population, $F_s$ is the scale factor which controls the amplification level of the differential variation or more simply the step size in the solution search process. The target vector is $\mathbf{p}_i$ and the mutant vector to be used for the evolution of the next generation is $\mathbf{m}_i$ $1 \leq i \leq p$. This is the standard version of the differential evolution algorithm [49] and is still widely used. Since a single difference of two randomly chosen candidates from the present generation is used to create the mutant candidate, the version is called DE/rand/1.
2.  DE/best/1: Instead of using the candidate vector, mutant vectors are generated using the same difference vector but now the scaled difference is added to the best candidate in the present generation, instead of the randomly chosen target vector.

$$\mathbf{m}_i(g+1) = \mathbf{best}(g) + F_s[\mathbf{p}_1(g) - \mathbf{p}_2(g)] \qquad (7)$$

where **best** is the best candidate based on fitness.
3.  DE/rand/2: The mutation operator is first used to generate a mutant vector by adding the scaled difference of two vectors to the third vector as,

$$\mathbf{m}_i(g+1) = \mathbf{p}_i(g) + F_s[\mathbf{p}_1(g) - \mathbf{p}_2(g)] + F_s[\mathbf{p}_3(g) - \mathbf{p}_4(g)] \qquad (8)$$

where $\mathbf{p}_3$ and $\mathbf{p}_4$ are two more randomly chosen candidates in the present generation.
4.  DE/best/2: The best candidate in the current generation is added to the scaled differences of two vectors as,

$$\mathbf{m}_i(g+1) = \mathbf{best}(g) + F_s[\mathbf{p}_1(g) - \mathbf{p}_2(g)] + F_s[\mathbf{p}_3(g) - \mathbf{p}_4(g)] \qquad (9)$$

5.  DE/current-to-rand/1: In this variation, the target vector is added to two scaled differences—one of them being the difference between a randomly chosen candidate

in the present generation and the target vector while the other is the difference between two randomly chosen candidates from the present generation.

$$\mathbf{m}_i(g+1) = \mathbf{p}_i(g) + F_s[\mathbf{p}_1(g) - \mathbf{p}_i(g)] + F_s[\mathbf{p}_2(g) - \mathbf{p}_3(g)] \tag{10}$$

6.  DE/current-to-best/1: The current-to-rand strategy is modified so that the scaled difference between a randomly chosen candidate and the target vector is replaced by the difference between the best candidate in the present generation and the target vector.

$$\mathbf{m}_i(g+1) = \mathbf{p}_i(g) + F_s[\mathbf{best}(g) - \mathbf{p}_i(g)] + F_s[\mathbf{p}_1(g) - \mathbf{p}_2(g)] \tag{11}$$

DE/rand/2 may result in better perturbation than strategies that use one difference vector [52]. DE/best/1 and DE/best/2 take advantage of the best candidate solution in the current population and have faster convergence towards the optimal solution [53]. DE/current-to-best/1 achieves a compromise between exploitation and exploration of the solution space, whereas DE/current-to-rand/1 has a rotation-invariant mutation strategy [54]. After mutation, the crossover operator is used to create trial candidates called trial vectors which then replace the target vector in the next generation based on their relative fitness. The trial vector is created using a bitwise or binomial crossover operator control by the crossover rate parameter $C_r$ from the interval [0, 1] as,

$$\mathbf{u}_{i,j}(g+1) = \begin{cases} \mathbf{m}_{i,j}(g+1) & \text{if } \eta \leq C_r \text{ or } j = \text{randint}(1, d) \\ \mathbf{p}_{i,j}(g) & \text{otherwise} \end{cases} \tag{12}$$

where, $\eta$ is a random number generated by using the uniform probability distribution in [0, 1]. The random integer *randint*(1, *d*) is an integer in the range [1, *d*] and is used to ensure that at least one mutant vector parameter is taken into account for constructing the trial vector. Since the binomial operator is used to effect bitwise crossover, the strategies are suffixed with a /bin, e.g., DE/rand/1/bin. The other commonly used crossover operator with differential evolution is the exponential operator. A starting point (bit) for crossover is chosen at random, and bit-wise elements of the trial vector either come from the mutant vector or the target vector depending on a series of Bernoulli experiments of probability $C_r$. The trial vector takes the mutant vectors bits until the Bernoulli experiment is unsuccessful for the first time or the trial vector is already complete. The remaining bits then come from the target vector. Together, the two parameters ($F_s$ and $C_r$) constitute the parameter set of the differential evolution algorithm. It is not difficult to see why DE is so popular—unlike other population-based optimization methods, DE only uses two parameters which for most problems are easy to tune.

To evolve a candidate, a tournament selection operator is used to compare the trial vector to the candidate vector using a one-to-one greedy selection criterion. The trial vector replaces the candidate vector in the new generation if it is better as measured by the fitness function $f$, otherwise the candidate vector in previous generation is retained.

$$\mathbf{p}_i(g+1) = \begin{cases} \mathbf{u}_i(g+1) & \text{if } f(\mathbf{u}_i(g+1)) > f(\mathbf{p}_i(g)) \\ \mathbf{p}_i(g) & \text{otherwise} \end{cases} \tag{13}$$

In addition, several updates of the basic strategy and its variations have been proposed with a view of speeding up convergence with good exploration in the initial generations. The other consideration is the self-tuning of the two parameters in the DE parameter set. The updates used in this paper are listed below.

7.  Composite DE (CoDE): This variant uses three different mutation strategies and three control parameter settings, combining them randomly to create trial vectors [55]. The three mutation strategies are rand/1/bin, rand/2/bin, and current-to-rand/1. The binomial crossover operator is not applied to current-to-rand/1. The choice of three control parameter settings will be discussed in Section 5. In a generation, for every

target vector, three trial vectors are created using each of the mutation strategies with one of the control parameter settings combined in a random manner. The best trial vector is then compared to the target vector and if better, enters the next generation.

8. jDE: The problem of effectively optimizing control parameter settings is addressed by a process of self-adapting these parameters within the DE process [56]. The control parameters $F_s$ and $C_r$ are adapted as,

$$F_s(g+1) = \begin{cases} F_l + rand_1 F_u & \text{if } rand_2 < \tau_1 \\ F_s(g) & \text{otherwise} \end{cases} \tag{14}$$

$$C_r(g+1) = \begin{cases} rand_3 & \text{if } rand_4 < \tau_2 \\ C_r(g) & \text{otherwise} \end{cases} \tag{15}$$

where $rand_1$, $rand_2$, $rand_3$, and $rand_4$ are uniform random variables in [0, 1], $\tau_1$ and $\tau_2$ are probabilities to adjust the control parameters, $F_l$ and $F_u$ are lower and upper bounds on the perturbation of the scale parameter in such a way that $F_{min} = F_l$ and $F_{max} = F_l + F_u$. The crossover rate parameter always takes values in [0, 1]. These adaptions are made prior to mutation i.e., creation of trial vectors and as a result they influence mutation, crossover, and selection operations going into the next generation.

9. JADE: JADE uses a differential mutation strategy called DE/current-to-pbest/1 and adapts the control parameters at every generation [57]. The current-to-pbest/1 mutation strategy is similar to current-to-best/1 except that instead of using the best candidate from the population **best**, to create a trial vector for a target vector, a vector **best**$^p$ is randomly chosen as one of the top $100p\%$ individuals in the current population with $p \in (0, 1]$. This ensures that second-best or third-best candidates also play a role in mutation for the next generation. The scale factor $F_s$ and the crossover rate $C_r$ are defined for each candidate in the population and a set of successful values is kept in an archive as evolution proceeds. Both parameters $(F_s)_i$ and $(C_r)_i$ for individual $\mathbf{p}_i$ are independently generated after every generation according to a Cauchy distribution for the scale factor and normal distribution for the crossover rate. The location parameter for the Cauchy distribution and the mean of the normal distribution are influenced by the previously found successful values of the parameters in the archive.

$$\mathbf{m}_i(g+1) = \mathbf{p}_i(g) + (F_s)_i[\mathbf{best}^p(g) - \mathbf{p}_i(g)] + (F_s)_i[\mathbf{p}_1(g) - \mathbf{p}_2(g)] \tag{16}$$

*4.2. Fitness Function*

A candidate $\mathbf{p}_i$ is decoded for the cluster number $c_i$ and cluster prototypes $v_i = \{v_{i1}, v_{i2}, \ldots v_{ici}\}$ and a preliminary assignment of data in the subset to clusters is performed as,

$$u_{ihk} = \frac{\left(\frac{1}{||v_{ih}-x_k||^2}\right)^{\frac{1}{m-1}}}{\sum\limits_{j=1}^{c_i}\left(\frac{1}{||v_{ij}-x_k||^2}\right)^{\frac{1}{m-1}}}, \ 1 \le h \le c_i, \ 1 \le k \le p \tag{17}$$

The *XB* index is defined as the ratio of the total variation $\sigma$ to the minimum separation $\lambda$ of the clusters [12]. The total variation is in fact the objective function with $m = 2$ and depends on the memberships and location of the cluster prototypes, while the minimum separation is the squared Euclidean distance between the closest cluster prototypes.

$$\begin{aligned} \sigma_i(U_i, V_i; X) &= \sum_{h=1}^{n_i} \sum_{k=1}^{p} u_{ihk}^2 ||v_{ih} - x_k||^2 \\ \lambda_i(V_i) &= \min_{h \neq j} ||v_{ih} - v_{ij}||^2 \\ XB_i &= \frac{\sigma(U_i, V_i; X)}{p\lambda(V_i)} \end{aligned} \tag{18}$$

When the clustering is compact and the number of clusters is evaluated correctly, the value of objective functional based $\sigma$ is minimized and the value of separation function $\lambda$

will be maximized, resulting in smaller values of the *XB* index. A cluster validity index $\Psi$ is defined in [58] as,

$$\Psi_i^2 = \frac{J_1(X)D_i(V_i)}{n_i J_i(U_i, V_i; X)} \tag{19}$$

where $J_i(U_i, V_i; X) = \sum_{h=1}^{c_i} \sum_{k=1}^{p} u_{ihk}||v_{ih} - x_k||$, $D_i(V_i) = \max_{h \neq j}||v_{ih} - v_{ij}||$ and $J_1(X)$ is constant for a given dataset and is calculated by setting $c_i = 1$ in the above equation (all datapoints are considered to be in the same cluster). It has been shown that good, compact clusters with the correct number of clusters identified will tend to maximize $\Psi$. The fitness for the candidate $\mathbf{p}_i$ in the population is evaluated for the differential evolution algorithm as a weighted sum of the two cluster validity indices,

$$f(\mathbf{p}_i) = \frac{a}{XB_i} + b\Psi_i^2 \tag{20}$$

where *a* and *b* are weights in [0, 1] that define the relative importance of the respective cluster validity index. In this paper, we choose $a = b = 0.5$.

### 4.3. Vector Representation

For the proposed method, a real number presentation is chosen to encode for cluster locations. All vectors in a population are the same length which corresponds to an *a priori* selected maximum number of clusters $c_{max}$. For a *d*-dimensional dataset, the vector will be of length $c_{max} \times d$. A binary masker for the vector of length $c_{max}$ is also used which controls for the activation of a cluster. For example, for $d = 3$ and $c_{max} = 6$, the real-number vector will be of length 18 and the masker will be of length 6 as shown in Figure 1 below.

**Vector**

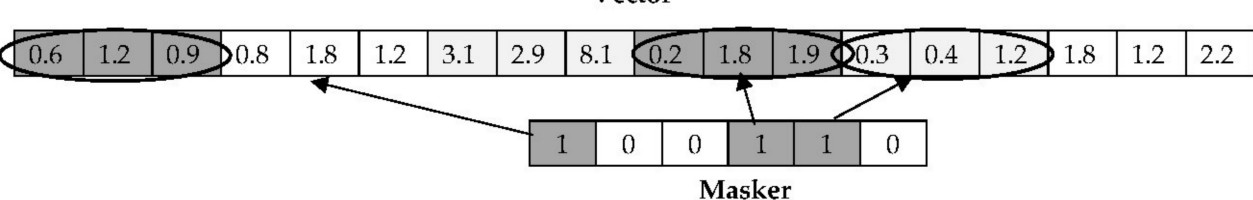

**Masker**

**Figure 1.** Example of an initial cluster prototype candidate vector and associated masker.

In the example above, the first, fourth and fifth cluster centers are activated using the marker which means the vector along with the marker represent a $c = 3$ partition with cluster centers $v_1 = (0.6, 1.2, 0.9)$, $v_2 = (0.2, 1.8, 1.9)$, and $v_3 = (0.3, 0.4, 1.2)$. The cluster centers not activated in the representation do not participate in fitness measurement. Every vector has an associated marker and although the vector participates in evolution, the marker does not. Instead, after every generation, the markers associated with each target vector in the new generation are updated with random binary numbers (simple mutation) except the marker associated with the best candidate in the new generation. If the masker encodes for $c < 2$, the masker is reinitialized until $c \geq 2$.

### 4.4. Multi-Population Parallel Differential Evolution with Elitism

The subset most similar in structure to the dataset is then provided to different subpopulations in the co-evolutionary algorithm. For each subpopulation, the *p* number of possible candidate solutions is encoded using the encoding strategy. Every subpopulation is also assigned a differential evolution (DE) strategy at random from a DE strategy set also explained in detail in this section. Each subpopulation is run for a total of *G* generations. An elite population is created after the first generation by sorting all individuals from every subpopulation and then selecting the top $p_e$ individuals to populate the elite population. Although the individuals in the elite population do not evolve, they participate in the differential strategy being implemented for every subpopulation in the next generation and

therefore contribute to the evolution of individuals in the subpopulations. At the culmination of every generation, the $p_e$ individuals of the elite population are mixed with the top $p_e$ number of individuals from the subpopulations and the top $p_e$ from the $2p_e$ combined population are selected as the subsequent elite population for the next generation. After $G$ generations, instead of selecting the top individual in the elite population as the optimal solution, we employ a consensus-based strategy. The $p_e$ individuals in the culminating elite population are decoded and grouped into bins based on the number of clusters (decoded via evolution). The most optimal individual from the largest bin is selected as the candidate solution and the cluster centers encoded by this individual are chosen as the initial cluster centers for FCM.

*4.5. Evaluation of Clustering Results*

After the initial cluster prototypes are evolved, they are then used as input to the FCM-AO algorithm for clustering. In Section 5, we present results from simulations conducted using both synthetic and benchmark datasets as well as a case study with exploratory data analysis from experimental data in Section 6. We compare the performance of various algorithms with our proposed method using several cluster validity indices from the literature. Cluster validity measures have been used in the literature for accessing the correctness of a partition. Since the Xie–Beni index *XB* and the $\Psi$ index are used as a combined measure of fitness, other cluster validity indices are used as a quantitative measure of comparison of various clustering techniques. The Bensaid Index $V_B$ is insensitive to the number of data points in a cluster [13]. The index is defined by the ratio of compactness of a cluster to the separation between clusters.

$$V_B = \sum_{k=1}^{c} \frac{\sum_{i=1}^{n} u_{ik}^m ||x_i - v_k||^2}{n_k \sum_{j=1}^{c} ||v_j - v_k||^2} \tag{21}$$

where $n_k$ is the fuzzy cardinality of cluster $k$ defined by $n_k = \sum_{i=1}^{n} u_{ik}$.

The final cluster prototypes are denoted by $\Upsilon = (v_1, v_2, \ldots, v_c)$. Note that the initial cluster centers are denoted by $v$ while the final cluster centers are denoted by $v$ to distinguish the two. The Bensaid index for a good partition is lower when compared to an inferior partition (one with an incorrect value of $c$ or incorrect cluster prototypes $v$). Tang et al. [14] introduced a punishing function which is the average distance between cluster centers-second term in the numerator in equation (22). This is used to counter the decreasing tendency of any cluster validity index as $c \to n$. The second term in the denominator is also a punishing function. The Tang index is denoted by $V_T$.

$$V_T = \sum_{k=1}^{c} \frac{\sum_{k=1}^{c} \sum_{i=1}^{n} u_{ik}^m ||x_i - v_k||^2 + \frac{1}{c(c-1)} \sum_{k=1}^{c} \sum_{\substack{j=1 \\ j \neq k}}^{c} ||v_k - v_j||^2}{\min_{j \neq k} ||v_j - v_k||^2 + \frac{1}{c}} \tag{22}$$

The modified Kwon index $V_{K2}$ proposed in [59] uses two terms in the numerator as punishing functions applied to eliminate the decreasing tendency as $c \to n$ and a dummy term for the stability of the index when $c \to n$ and $m \to \infty$, respectively. The second term of the denominator is an ad hoc punishing function used to strengthen the numerical stability as $m \to \infty$. The last term of the denominator is another ad hoc punishing function used to strengthen the numerical stability as $c \to n$.

$$V_{K2} = \frac{\frac{n-c+1}{2} \left[ \left(\frac{c}{c-1}\right)^{\sqrt{2}} \sum_{k=1}^{c} \sum_{i=1}^{n} u_{ik}^{2^{\sqrt{m/2}}} ||x_i - v_k||^2 + \frac{\sum_{j=1}^{c} ||v_j - \bar{v}||^2}{\max_j ||v_j - \bar{v}||^2} + \frac{nc}{(n-c+1)^2} \right]}{\min_{j \neq k} ||v_j - v_k||^2 + \frac{1}{c} + \frac{1}{c^{m-1}}} \tag{23}$$

where, $\overline{v} = \frac{1}{n} \sum\limits_{j=1}^{n} x_j$ is the mean cluster prototype when $c = 1$.

Ren index $V_R$ [60] is an improvement on the Bensaid index $V_B$. The numerator represents the compactness of the cluster $C_k$, where $n_k$ is its fuzzy cardinality. Its second item, an introduced punishing function, denotes the distance from the cluster prototype of the $k$th cluster to the average of all cluster prototypes, which can eliminate the monotonically decreasing tendency as the number of clusters increases to $n$. The denominator represents the mean distance from the $k$th cluster prototype to all other cluster prototypes, which is a measure of intercluster separation. The ratio of the numerator and the denominator represents the clustering effect of the $k$th cluster. The clustering validity index is defined as the sum of the clustering effect (the ratio) of all clusters. The smaller the index is, the better the clustering effect of the dataset is, and the corresponding $c$ to the minimum value is the optimal number of clusters.

$$V_R = \sum_{k=1}^{c} \frac{\frac{1}{n_k} \sum_{i=1}^{n} u_{ik}^m ||x_i - v_k||^2 + \frac{1}{c} ||v_k - \overline{v}||^2}{\frac{1}{c-1} \sum_{j=1}^{c} ||v_j - v_k||^2} \tag{24}$$

The cooperative multi-population differential evolution algorithm with elitism is presented below as Algorithm 3 and will be referred to as parallel coevolution.

---

**Algorithm 3:** Parallel Coevolution

---

**Input:** Best Subset $P_{best\_subset}$, B number of DE strategies DE = {$DE_1$, $DE_2$, ... $DE_B$}, fitness criterion f, total generations G, number of subpopulations P, number of individuals in each subpopulation p, number of individuals in the elite population $p_e$.

1.   Create P subpopulations S = {$S_1$, $S_2$, ... $S_P$} by randomly populating p candidates for each subpopulation
2.   **For** i = 1 to P
3.   rand = random number between 1 and B
4.   $DES_i$ = Assign strategy $DE_{rand}$ to $S_i$
5.   **End**
6.   Elite population E = Ø
7.   **For** gen = 1 to G
8.   **For** i = 1 to S
9.   $S_i = S_i$ U E
10.  Evolve subpopulation $P_i$ using strategy $DES_i$
11.  **End**
12.  TG = Sort all p x P individuals in decreasing order of fitness using criterion f
13.  Transient elite population TE = [Top $p_e$ individuals from TG]
14.  TTE = TE U E
15.  E = [Top $p_e$ individuals from TTE]
16.  **End**
17.  //Consensus-Based Selection//
18.  Create bins C = {$C_1$, $C_2$, ... $C_{max\_n}$}
19.  **For** j = 1 to $p_e$
20.  Decode $E_j$ = [clusters_centers$_j$, fitness$_j$, number of clusters $E_{nj}$]
21.  Assign $E_j$ to bin $C_k$ where k = $E_{nj}$
22.  **End**
23.  largest_bin = {$C_1$}
24.  **For** j = 2 to max_n
25.  **If** size($C_j$) >= size(largest_bin)
26.  largest_bin = largest_bin U $C_j$
27.  **End**
28.  **End**

---

**Output:** Select fittest individual from largest_bin, use cluster_centers from this individual as initial cluster centers

---

## 5. Simulation Results

The proposed method is compared to some standard algorithms from the literature and their parameters used in this study are listed below.

1. The canonical FCM or FCM-AO is implemented with $m = 2$ and membership convergence of $\varepsilon = 0.001$. The value of $c$ is varied from $c = 2$ to $c = \sqrt{n}$. To compare with the proposed method, FCM-AO is run parallelly with different initializations for each value of $c$.

2. A single differential evolution-based FCM called automatic differential evolution-based fuzzy clustering (ADEFC) in [9] is run for $G = 100$ generations with a population size of 40. The crossover probability $C_r = 0.8$ and the scale factor $F_s = 0.5$. The vector representation of DE-FCM is the same as that used in the proposed method.

3. The PSO-V variant of the particle swarm optimization-based FCM presented in [61] is based on FCM-AO when the cluster prototypes are randomly initialized. The datasets used are run for 50 particles (50 different cluster prototype initializations) for 1000 iterations. The acceleration parameters $a_1 = a_2 = 1.5$ and the maximum velocity is chosen as $\Delta y_{\max} = 0.25$. The number of clusters are varied from $c = 2$ to $c = \sqrt{n}$ and all implementations are run parallelly.

4. The Kernel-based fuzzy c-means (KFCM) using genetic algorithms (GA) is named GAKFCM [32]. It uses a standard RBF kernel and GA parameters: population size of 50, max iterations of 500, crossover and mutation probabilities of 0.6 and 0.05, respectively. The FCM parameters are the same as those used for FCM-AO. The number of clusters is varied from $c = 2$ to $c = \sqrt{n}$ and all implementations are run parallelly.

5. The entropy weighted-FCM (or EwFCM) as presented in [30] is implemented with an entropy threshold parameter set to 0.05, the maximum number of cycles of 150. The number of clusters is varied from $c = 2$ to $c = \sqrt{n}$ and all implementations are run parallelly.

We do not present these algorithms in detail here and the reader is referred to the original papers. The proposed algorithm will henceforth be referred to as Cooperative DE-FCM or CDE-FCM. The parameters are SP = 10, number of subpopulations = 9 (one for each instance of DE), generations $G = 20$, number of candidate vectors in each subpopulation $p = 15$. The individual parameter settings of the DE variants are listed in Tables 1 and 2. We compared the performance of these algorithms using cluster validity measures listed in Section 4.5 on two synthetic datasets and four datasets from the UCI ML repository [62]. All algorithms are implemented using MATLAB 2022a on an Intel® Core™ i7-8650U CPU with 8 cores at 1.90 GHz.

**Table 1.** Parameter settings for DE variants 1–6.

|       | DE/rand/1/bin | DE/best/1/bin | DE/rand/2/bin | DE/best/2/bin | DE/current-to-rand/1 | DE/current-to-best/1 |
|-------|---------------|---------------|---------------|---------------|----------------------|----------------------|
| $F_s$ | 1.0           | 1.0           | 0.5           | 0.5           | 0.8                  | 0.8                  |
| $C_r$ | 0.9           | 0.1           | 0.9           | 0.9           | 0.1                  | 0.1                  |

**Table 2.** Parameter settings for DE variants 7–9.

|  | CoDE | jDE | JADE |
|---|---|---|---|
| DE Set | DE/rand/1/bin<br>DE/rand/2/bin<br>DE/current-to-rand/1 |  |  |
| $[F_s, C_r]$ | [1.0, 0.1]<br>[1.0, 0.9]<br>[0.8, 0.2] |  |  |
| $F_l$ |  | 0.1 |  |
| $F_u$ |  | 0.9 |  |
| $\tau_1$ |  | 0.1 |  |
| $\tau_2$ |  | 0.1 |  |
| $C_r$ |  | 0.5 |  |
| Parameter adaption rate, $c$ |  |  | 0.1 |
| % Top best, $p$ |  |  | 5 |
| Initial $F_s$ |  |  | 0.5 |
| Initial $C_r$ |  |  | 0.5 |

**Data1** is a 900-point dataset in two dimensions. There are three well-separated and compact clusters with 300 data points in each cluster. The clusters are generated using normal distributions centered at (1,1), (3,3), and (4,5). The dataset is shown in Figure 2. The three-cluster FCM-AO partition is shown in Figure 3 and the three-cluster CDE-FCM after cluster prototype initialization is shown in Figure 4. The quality of the CDE-FCM can be seen to be marginally better than randomly initialized FCM-AO.

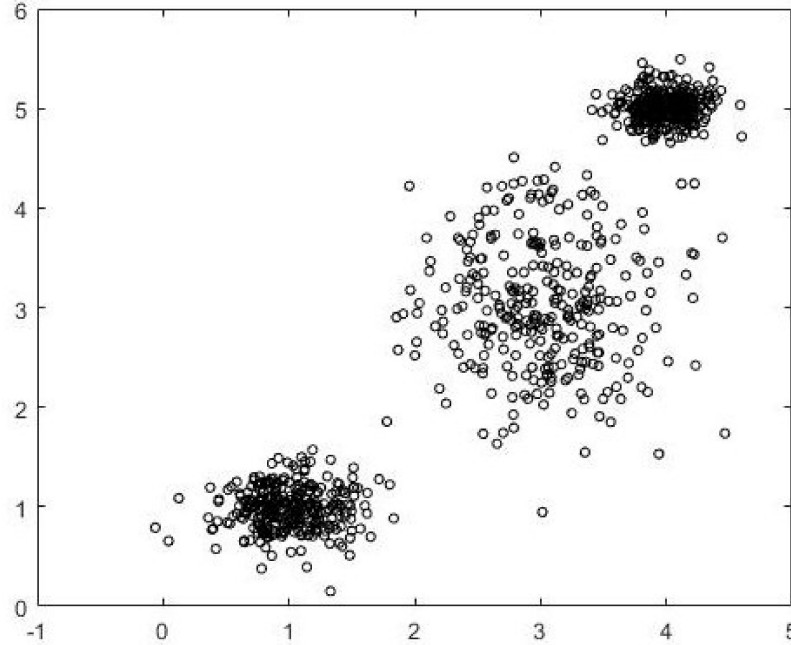

**Figure 2.** Data1 ($n$ = 900, $c$ = 3, $H$ = 0.9727).

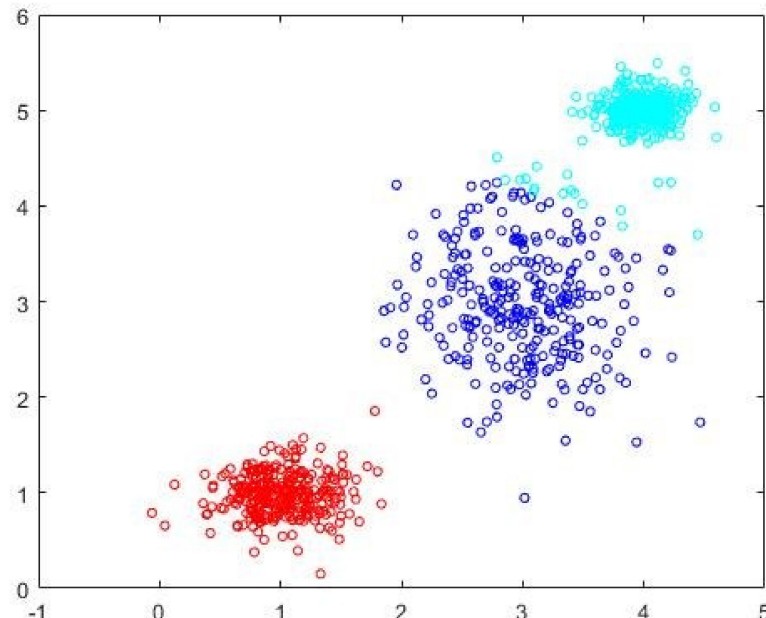

**Figure 3.** FCM-AO with randomly initialized cluster prototypes for *c* = 3.

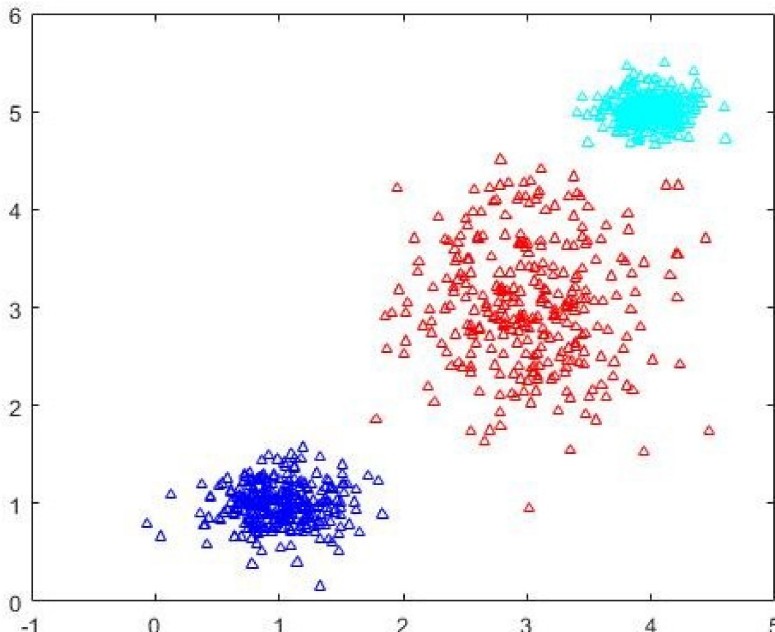

**Figure 4.** CDE-FCM with evolved initial cluster prototypes for *c* = 3.

FCM-AO, PSO-V, GAKFCM, and EwFCM are run for *c* = 2 to *c* = 30. For ADEFC and CDE-FCM the vector representation is of length 60 with *d* = 2 and $c_{max}$ = 30. The run-time parameters are chosen such as the convergence criteria for all algorithms are roughly the same. For comparison, the run-time of the different algorithms is presented in Table 3. As can be seen, there is not much to be gained by using sophisticated schemes to address the problems of prototype initialization and unknown number of clusters for a small *well-separated* dataset as Data1. The naïve version of FCM-AO with random initializations works as well as any other scheme and in a fraction of the time.

**Table 3.** Runtime of algorithms for **Data1**.

|  | FCM-AO | ADEFC | PSO-V | GAKFCM | EwFCM | CDE-FCM |
|---|---|---|---|---|---|---|
| Run time (s) | 2.85 | 8.39 | 5.67 | 8.45 | 4.11 | 8.45 |

**Data2** is a 2000-point dataset in two dimensions. Unlike **Data1**, the separation between the clusters is not well defined. The data is generated using three normal distributions centered at (3,2), (5.5,5.5) and (8.5,6). The intercluster variance is greater than that in **Data1**. The dataset is shown in Figure 5. FCM-AO, PSO-V, GAKFCM and EwFCM are run for $c = 2$ to $c = 45$. For ADEFC and CDE-FCM, the vector representation is of length 90 with $d = 2$ and $c_{max} = 45$. FCM-AO, GAKFCM, and EwFCM identify $c = 5$ as the most-optimal cluster. On the other hand, the evolutionary algorithm-based approaches, viz. PSO-V, ADEFC and the proposed CDE-FCM correctly identify the $c = 3$ partition. In fact, the terminating elite population in CDE-FCM has 22% individuals that encode for $c = 3$ and 10% that encode for $c = 5$. The best subset of cardinality 200 as identified by Algorithm 2 is shown in Figure 6 and the initial cluster prototypes evolved by the CDE (Algorithm 3) are shown in Figure 7. The $c = 5$ partition as identified by FCM-AO as the most optimal partition is shown in Figure 8 and the $c = 3$ partition as identified by FCM-AO is shown in Figure 9. The correct $c = 3$ partition as identified by CDE-FCM with evolved initial prototypes is shown in Figure 10. By simple visual inspection, it can be said that the quality of the partition in Figure 10 is better than that in Figure 9 (same $c = 3$). The fact that FCM-AO with randomly initialized cluster prototypes identifies $c = 5$ as better than $c = 3$ on the basis of all cluster validity measures shows the promise of the proposed method. After the initial cluster prototypes are identified by Algorithm 3, the FCM-AO converged on an average in 4.5 iterations over 10 implementations. This is in comparison to the naïve FCM-AO with random initialization which took at least 25 iterations for an average of 28 iterations with the same convergence criterion of $\varepsilon = 0.001$.

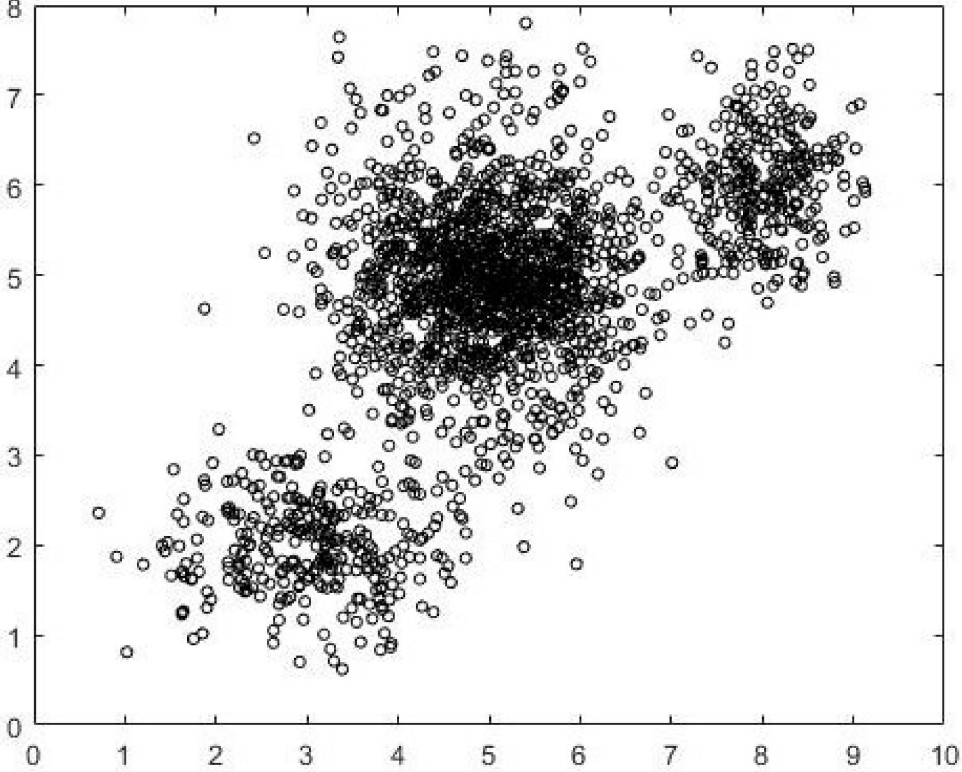

**Figure 5. Data2** ($n = 2000$, $c = 3$, $H = 0.9184$).

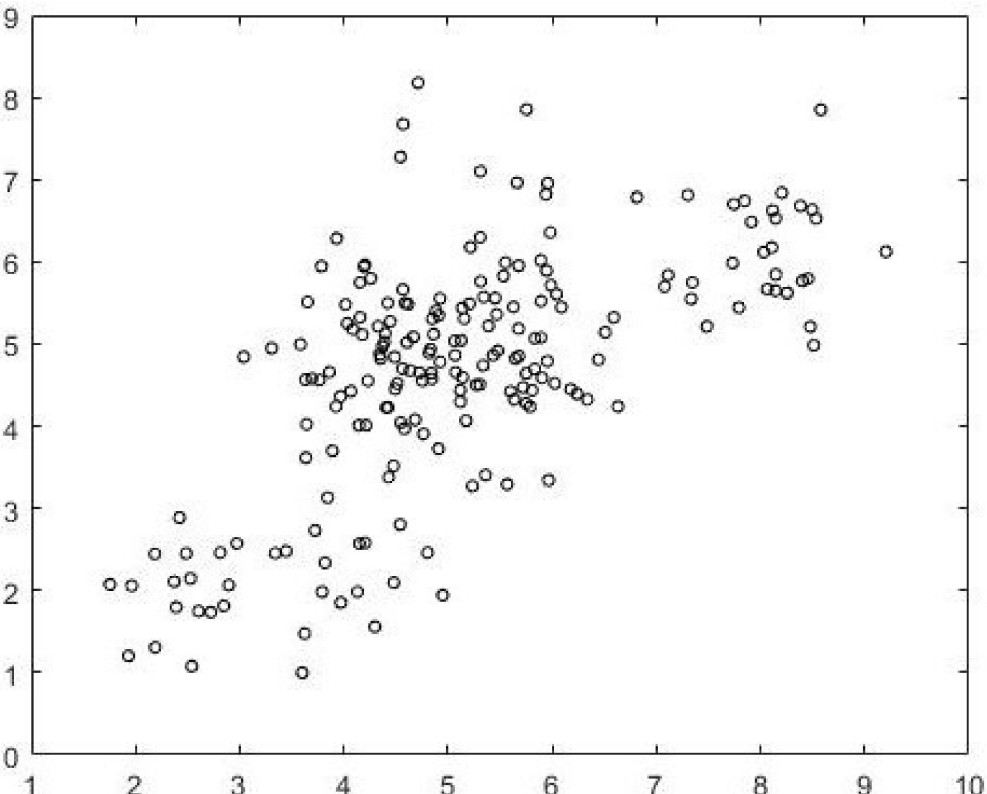

**Figure 6.** Best subset of **Data2** using sparse sampling ($n = 200$, $H = 0.9025$).

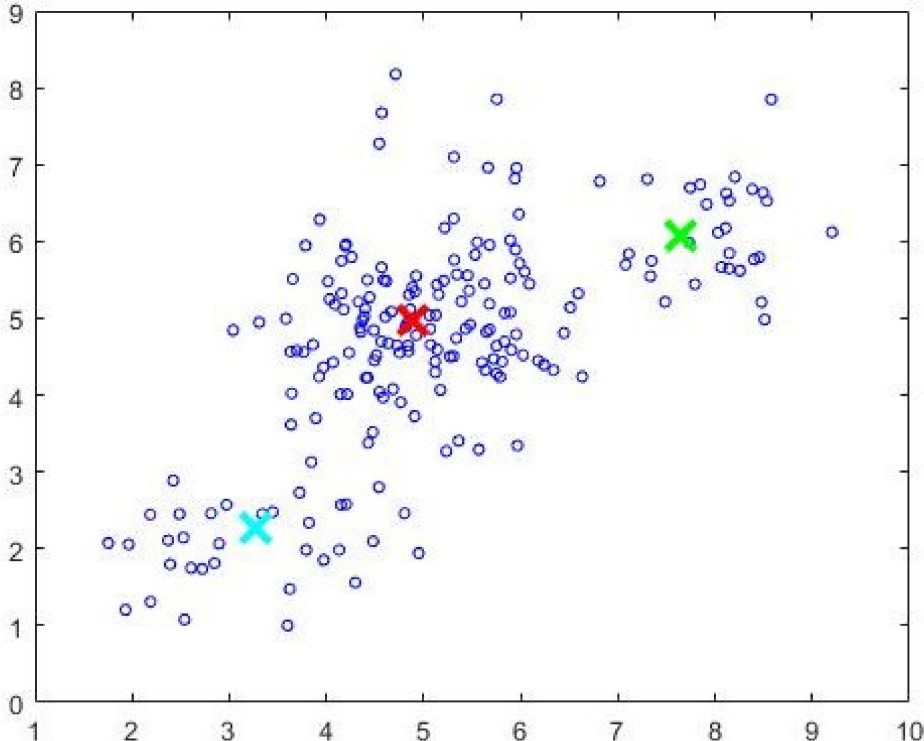

**Figure 7.** Initial cluster prototype as calculated by CDE on best subset of **Data2.** The Xs mark the initial cluster prototype location.

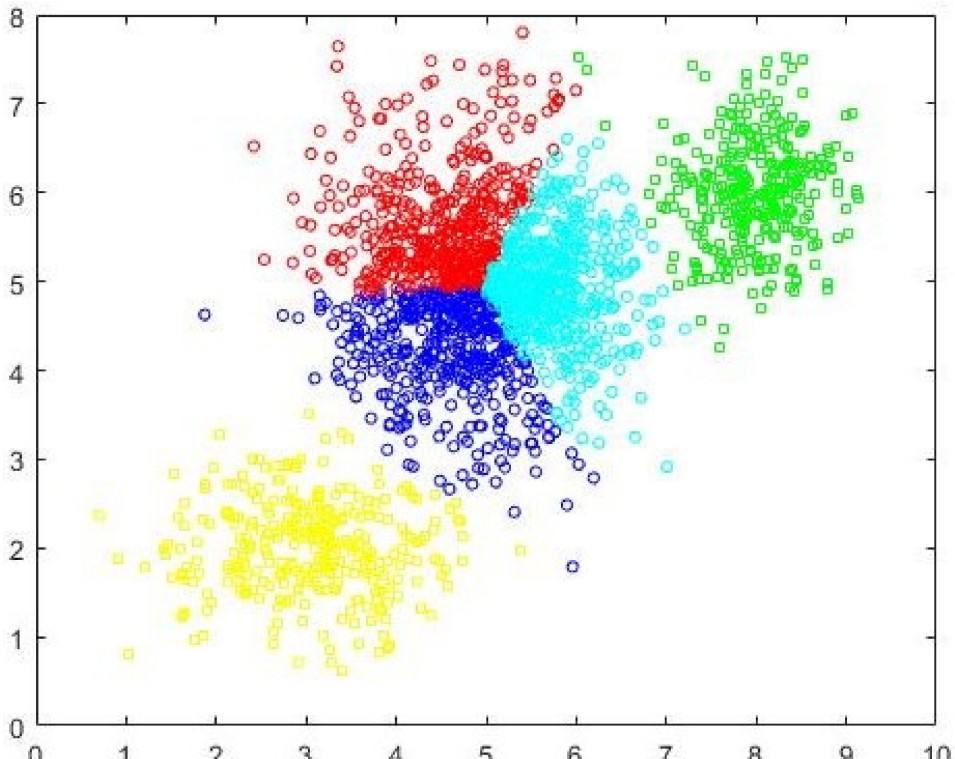

**Figure 8.** Best partition by FCM-AO, *c* = 5 with randomly initialized cluster prototypes.

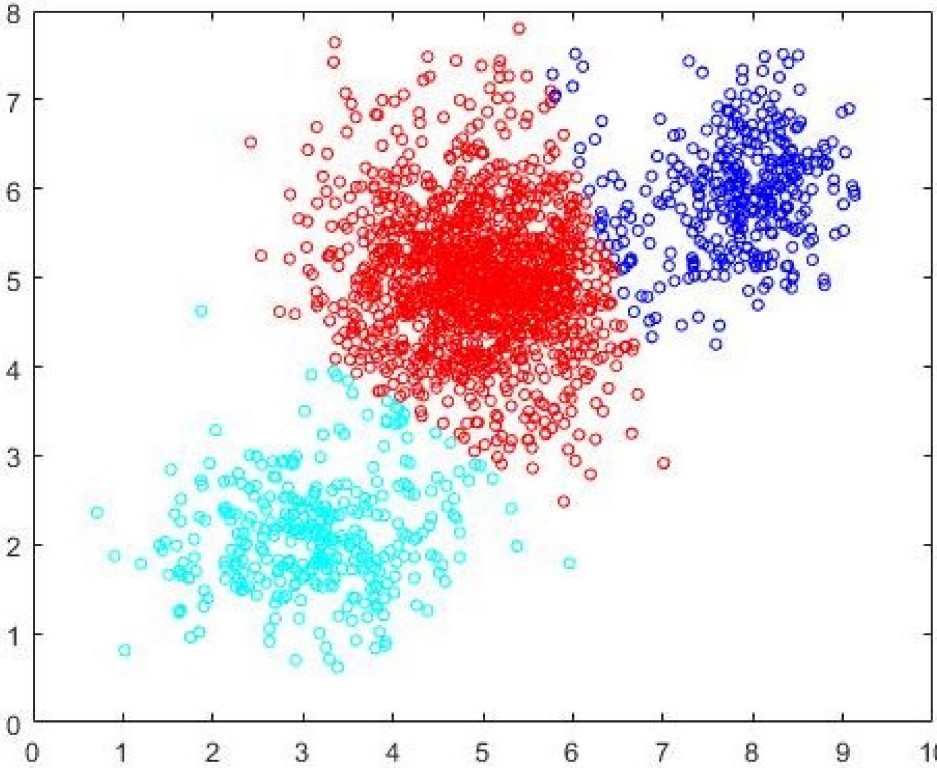

**Figure 9.** Partition by FCM-AO, *c* = 3 with randomly initialized cluster prototypes.

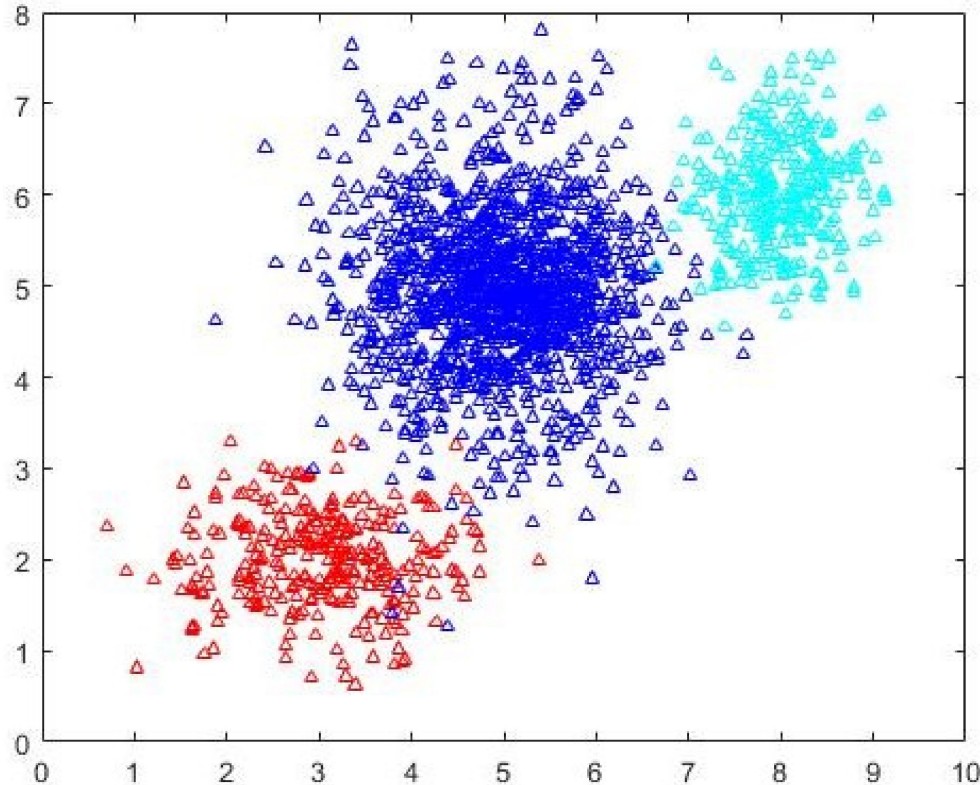

**Figure 10.** Partition by CDE-FCM, *c* = 3 with initial cluster prototypes shown in Figure 7.

**Iris** data consists of *n* = 150 datapoints divided into three types of Iris flowers-Setosa, Virginica, and Versicolor with 50 samples in each class. Each sample has four associated features: sepal length, petal length, sepal width, and petal width. FCM-AO, PSO-V, GAKFCM and EwFCM are run for *c* = 2 to *c* = 12. For ADEFC and CDE-FCM the vector representation is of length 48 with *d* = 4 and $c_{max}$ = 12. The performances of different algorithms as measured by the cluster validity criteria are listed in Table 4. Two classes (Versicolor and Virginica) are known to be linearly inseparable from each other although Setosa is linearly separable from the other classes [63] and therefore many algorithms identify the *c* = 2 solution as the most optimal one. Almost 20% of the candidate vectors in the terminating elite population of CDE-FCM encoded for *c* = 2 while 18% encoded for *c* = 3 with an equal number encoding for *c* = 9. The cluster prototypes identified by CDE-FCM in almost all candidates (>85%) in the terminating elite population are almost optimal, and a further application of FCM-AO converges on average in 4.1 iterations over 10 independent runs of the algorithm. On the other hand, the best-performing instance of the naïve FCM-AO with random initializations took 22 iterations.

**Table 4.** Performance of algorithms on Iris data.

|  | *c* | $V_B$ | $V_T$ | $V_{K2}$ | $V_R$ |
|---|---|---|---|---|---|
| FCM-AO | 2 | 0.2302 | 0.0682 | 8.9245 | 0.4789 |
|  | 3 | 0.1349 | 0.1371 | 19.235 | 0.5529 |
|  | 9 | 0.0284 | 0.3672 | 67.224 | 0.5849 |
| ADEFC | 9 | 0.0296 | 0.3619 | 71.344 | 0.5849 |
| PSO-V | 2 | 0.2311 | 0.0671 | 8.9382 | 0.4762 |
|  | 3 | 0.1245 | 0.1371 | 19.235 | 0.5529 |
|  | 9 | 0.0278 | 0.3774 | 67.334 | 0.5872 |
| GAKFCM | 2 | 0.2372 | 0.0680 | 8.9263 | 0.4821 |
|  | 3 | 0.1262 | 0.1371 | 18.936 | 0.5587 |
| EwFCM | 3 | 0.1263 | 0.1362 | 19.292 | 0.5529 |
|  | 4 | 0.1021 | 0.1942 | 32.039 | 0.6232 |
|  | 9 | 0.0287 | 0.3672 | 71.348 | 0.5869 |
| CDE-FCM | 2 | 0.2235 | 0.0590 | 8.4385 | 0.4741 |

**Cancer** data has 683 datapoints in two classes (malignant and benign). The data has 9 features (attributes)—clump thickness, cell size uniformity, cell shape uniformity, single epithelial cell size, bare nuclei, bland chromatin, normal nucleoli, and mitoses. There are 440 instances belonging to the benign cluster and 243 instances in the malignant cluster. The cancer dataset is known to be linearly inseparable and it is often difficult for clustering algorithms to achieve high levels of accuracy with this dataset. FCM-AO, PSO-V, GAKFCM and EwFCM are run for $c = 2$ to $c = 26$. For ADEFC and CDE-FCM the vector representation is of length 234 with $d = 9$ and $c_{max} = 26$. The performances of different algorithms as measured by the cluster validity criteria are listed in Table 5. Little more than 18% of the candidate vectors in the terminating elite population of CDE-FCM encoded for $c = 2$ while approximately 15% of candidate vectors encoded for $c = 8$. The cluster prototypes identified by CDE-FCM in 72% of the candidates in the terminating elite population are almost optimal, and a further application of FCM-AO converges on average in 8.5 iterations over 10 independent runs of the algorithm. On the other hand, the best-performing instance of the naïve FCM-AO with random initializations took 31 iterations.

**Table 5.** Performance of algorithms on cancer data.

|  | $c$ | $V_B$ | $V_T$ | $V_{K2}$ | $V_R$ |
|---|---|---|---|---|---|
| FCM-AO | 2 | 0.1233 | 0.0121 | 12.897 | 0.2868 |
|  | 8 | 0.0129 | 0.6792 | 128.329 | 0.3862 |
| ADEFC | 2 | 0.1234 | 0.0123 | 12.775 | 0.2854 |
| PSO-V | 2 | 0.1293 | 0.0129 | 12.292 | 0.2841 |
|  | 8 | 0.0137 | 0.6822 | 137.850 | 0.3922 |
| GAKFCM | 2 | 0.1224 | 0.0174 | 12.325 | 0.2833 |
|  | 8 | 0.0193 | 0.6891 | 129.825 | 0.3851 |
| EwFCM | 2 | 0.1239 | 0.0132 | 12.457 | 0.2857 |
|  | 8 | 0.0174 | 0.6891 | 135.839 | 0.3823 |
| CDE-FCM | 2 | 0.1197 | 0.0121 | 12.775 | 0.2803 |

**Glass** data has 214 different glass samples with 9 features (refractive index, weight percent of corresponding oxide of Na, Mg, Al, Si, K, Ca, Ba, and Fe). There are 6 different types of glasses—building window (float), building window (non-float), vehicle window (float), container glass, tableware glass and headlamp glass. FCM-AO, PSO-V, GAKFCM and EwFCM are run for $c = 2$ to $c = 15$. For ADEFC and CDE-FCM the vector representation is of length 135 with $d = 9$ and $c_{max} = 15$. The performances of different algorithms as measured by cluster validity criteria are listed in Table 6. Almost all cluster validity indices agreed, and all algorithms produced very similar partitions at either $c = 5$, $c = 6$, and $c = 8$.

**Table 6.** Performance of algorithms on glass data.

|  | $c$ | $V_B$ | $V_T$ | $V_{K2}$ | $V_R$ |
|---|---|---|---|---|---|
| FCM-AO | 5 | 0.8251 | 0.1925 | 12.854 | 3.332 |
|  | 6 | 0.8149 | 0.1786 | 12.779 | 3.297 |
| ADEFC | 5 | 0.8135 | 0.1883 | 12.775 | 3.472 |
| PSO-V | 5 | 0.8238 | 0.1937 | 12.854 | 3.974 |
|  | 8 | 0.9244 | 0.2215 | 32.975 | 8.773 |
| GAKFCM | 5 | 0.8188 | 0.1875 | 12.695 | 3.224 |
|  | 6 | 0.8275 | 0.1786 | 12.702 | 3.133 |
| EwFCM | 6 | 0.8175 | 0.1775 | 12.972 | 3.297 |
|  | 8 | 0.8924 | 0.2232 | 37.395 | 9.725 |
| CDE-FCM | 6 | 0.8077 | 0.1760 | 12.715 | 3.109 |

**Wine** dataset has 178 samples of wine differentiated by 13 attributes—percent contents of alcohol, malic acid, ash, magnesium, total phenols, flavonoids, nonflavonoid phenols, proanthocyanins, alkalinity of ash, color intensity, OD280/OD315 of diluted wines (protein content), and proline content. The wines are categorized into three classes (red, white, and rosé) with 59 instances in the first cluster, 71 in the second, and the rest in the third cluster. FCM-AO, PSO-V, GAKFCM, and EwFCM are run for $c = 2$ to $c = 13$. For ADEFC and CDE-FCM the vector representation is of length 169 with $d = 13$ and $c_{max} = 13$. The performances of different algorithms as measured by cluster validity criteria are listed in Table 7. The clusters are well-defined, and all the cluster validity measures reach their minimum at either $c = 3$ or $c = 4$.

**Table 7.** Performance of algorithms on Wine data.

|  | $c$ | $V_B$ | $V_T$ | $V_{K2}$ | $V_R$ |
|---|---|---|---|---|---|
| FCM-AO | 3 | 1.025 | 0.0997 | 8.726 | 2.970 |
| ADEFC | 4 | 1.133 | 0.1029 | 12.339 | 3.892 |
| PSO-V | 3 | 1.025 | 0.0987 | 8.697 | 2.835 |
| GAKFCM | 3 | 1.025 | 0.0997 | 8.645 | 2.754 |
| EwFCM | 3 | 1.175 | 0.0984 | 8.772 | 2.663 |
| CDE-FCM | 3 | 1.025 | 0.0984 | 8.597 | 2.754 |

**Wine Quality** dataset has 1898 samples of wines divided into 11 classes (quality scores ranging from 0 to 10). The data are defined over 11 attributes—fixed acidity, volatile acidity, citric acid, residual sugar, chlorides, free $SO_2$, total $SO_2$, density, pH, sulphates, and alcohol content. This is the largest of the UCI datasets tested for this work both in terms of cardinality and dimensions. FCM-AO, PSO-V, GAKFCM and EwFCM are run for $c = 2$ to $c = 45$. For ADEFC and CDE-FCM the vector representation is of length 495 with $d = 11$ and $c_{max} = 45$. The performances of different algorithms as measured by cluster validity criteria are listed in Table 8. FCM-AO, PSO-V, GAKFCM, and EwFCM identify at least 8 of the classes with moderately high accuracy. Accuracy decreases for $c = 11$ as is evident from the cluster validity measures. ADEFC identifies $c = 10$ clusters, with an accuracy below 90% while the proposed CDE-FCM identifies 12 clusters with initial cluster centers almost optimal for 8 of the real classes. The FCM-AO implemented with cluster prototypes identified by CDE-FCM converges on an average of 12 iterations over 10 independent runs, while FCM-AO randomly initialized converges after an average of 33 iterations for $c = 8$ and 37 iterations for $c = 11$.

**Table 8.** Performance of algorithms on wine quality data.

|  | $c$ | $V_B$ | $V_T$ | $V_{K2}$ | $V_R$ |
|---|---|---|---|---|---|
| FCM-AO | 8 | 8.775 | 0.3721 | 72.456 | 5.978 |
|  | 10 | 13.942 | 0.9885 | 125.814 | 8.775 |
|  | 11 | 22.857 | 0.9925 | 216.380 | 12.394 |
|  | 12 | 23.617 | 1.0225 | 229.773 | 12.945 |
| ADEFC | 10 | 14.332 | 0.9885 | 120.035 | 8.674 |
| PSO-V | 8 | 8.456 | 0.3389 | 70.990 | 5.825 |
|  | 10 | 13.880 | 0.9727 | 118.336 | 8.650 |
|  | 11 | 21.352 | 0.9880 | 207.350 | 12.218 |
|  | 12 | 21.445 | 1.0225 | 219.872 | 12.945 |
| GAKFCM | 8 | 8.356 | 0.3392 | 71.855 | 5.978 |
|  | 11 | 21.335 | 0.9891 | 208.392 | 12.218 |
| EwFCM | 8 | 8.470 | 0.3391 | 70.990 | 5.824 |
|  | 10 | 13.442 | 0.9738 | 115.275 | 8.458 |
|  | 11 | 21.832 | 0.9890 | 208.360 | 12.394 |
|  | 12 | 21.975 | 1.0225 | 220.825 | 12.945 |
| CDE-FCM | 12 | 21.329 | 1.0225 | 217.375 | 12.885 |

## 6. Case Study—Rolling Bearing Fault Analysis

Rolling element bearings are commonly used in supporting rotor components and assemblies in rotating machinery. Bearing defects can lead to undesirable vibrations, noise,

or machine failure. Bearing fault diagnosis has been a subject of great importance in machine condition monitoring, predictive maintenance, and machine failure prevention and analysis [64]. Bearings conditions are presented in [65]. Techniques used in fault severity evaluation in rolling bearings are reviewed and discussed in [66]. The review is mainly focused on data-driven approaches such as signal processing for extracting the fault signatures associated with the fault degradation, and the approaches that are used to identify degradation patterns. Modern predictive maintenance techniques are increasingly adopting data analysis techniques such as pattern recognition and machine learning for bearing fault diagnosis. In this case study using the proposed cluster initialization method, wavelet analysis is used to process vibration signals from three bearing cases—no fault, inner race fault, and ball fault, under varying rotating speeds.

A schematic of the experimental setup is shown in Figure 11. The faults are introduced as a single rough surface spot simulating pitting wear and are created by using a small grinder as shown in Figure 12. The rotor is run at 10 different speeds (from 500 rpm to 1400 rpm) in increments of 100 rpm. At each speed level, vibration signals are acquired using a PCB accelerometer (model PCB 302A) which is mounted on the outboard test bearing as indicated in Figure 10. The sampling rate is 10,000 samples/s. A radial load of about 2000 lb was kept constant during all tests. A small unbalance mass is added to the rotor to introduce sustained periodic vibration excitation.

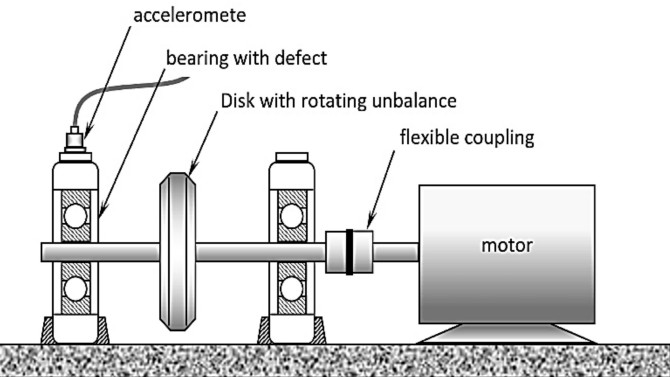

**Figure 11.** Experimental setup.

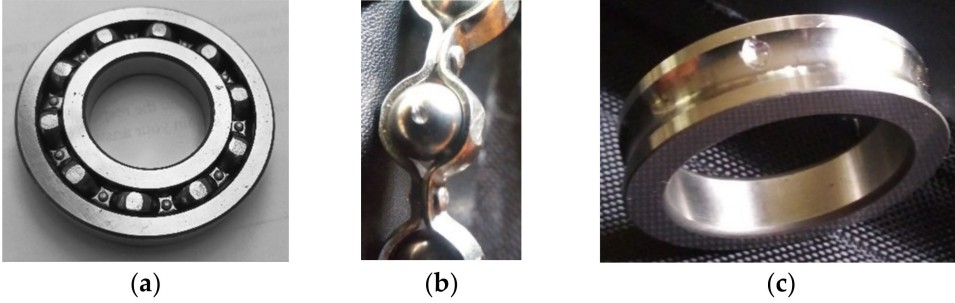

**Figure 12.** (**a**) Ball bearing used in the test, (**b**) ball fault, (**c**) inner race fault.

Three sets of samples are collected, each for 0.5 s. The analysis is performed for each of the three sets, followed by sets obtained using a 50% overlap between the non-overlapping sets resulting in five sample sets (3 non-overlapping sets and 2 overlapping sets) for a single operating condition—speed and type of fault (10 different speeds and 3 fault conditions including no fault). The raw time series is analyzed for statistical features such as mean, variance, standard deviation, skewness, and kurtosis. The times series are also subject to fast Fourier transform (FFT) and continuous wavelet transform (CWT) techniques. Figure 13 shows two sample raw vibration time series and their corresponding frequency spectrum obtained using FFT. After experimenting with several wavelet transform functions, the 'Mexican hat' and the Coiflet wavelet functions were used. The CWT implementation was

performed using MATLAB's wavelet toolbox. MATLAB's CWT can be considered as a filter that scales a mother wavelet function along the time axis. At each scale, the CWT function will superimpose the scaled mother wavelet wave form over a segment of the signal under analysis. The similarities and differences between the form of the wavelet wave and the signal being analyzed are determined by CWT as,

$$C(a, T) = \int \frac{1}{\sqrt{a}} \psi(t) \left( \frac{t - T}{a} \right) x(t) dt \qquad (25)$$

where $\psi(t)$ represents the CWT mother wavelet function which is shifted in time by $T$ and dilated or contracted by a factor and then correlated with the vibration signal represented by $x(t)$.

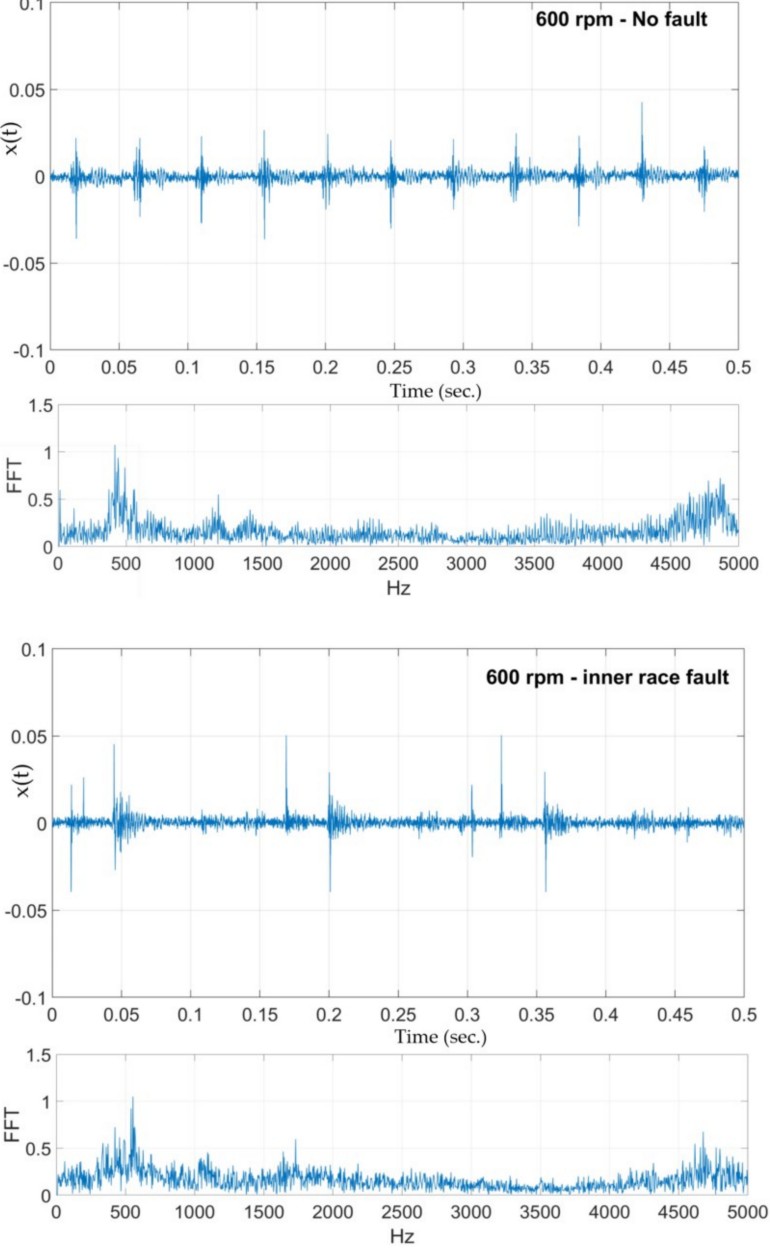

**Figure 13.** Vibration time signatures and corresponding FFT spectrum for two cases.

All feature attributes are scaled and normalized across the column before clustering. The datasets are unlike others used in the paper—high dimensional data with relatively low

cardinality. The idea is to see if the fault conditions can themselves be or if any combination of operating speed and fault condition can be partitioned. The datasets are described below:

**BearingData1** is three non-overlapping segments with 16 FFT averages, $n = 90$, $d = 16$

**BearingData2** is three non-overlapping segments with 64 Mexican Hat averaged wavelet coefficients, $n = 90$, $d = 64$

**BearingData3** is three non-overlapping segments with 64 Coiflet-averaged wavelet coefficients, $n = 90$, $d = 64$

**BearingData4** is three non-overlapping segments with kurtosis, skewness, RMS, and crest factors for 10 wavelet approximations and 10 wavelet details, $n = 90$, $d = 80$.

We also use a low dimensional data with non-overlapping and overlapping regions of the raw data (50% overlap). **BearingData5** is three non-overlapping and two overlapping segments with four statistical features of the raw time signal—mean, skewness, standard deviation, and kurtosis, $n = 150$, $d = 4$.

The performance of the proposed method is compared with FCM-AO with random initialization of cluster prototypes using the four cluster validity indices. FCM-AO is implemented for $c = 2$ to $c = 10$. The vector representation used in CDE-FCM is $d \times 10$. The results are tabulated in Table 9.

**Table 9.** Performance of FCM-AO and CDE-FCM for five bearing fault datasets.

| | $c$ | $V_B$ | $V_T$ | $V_{K2}$ | $V_R$ |
|---|---|---|---|---|---|
| **BearingData1**, $n = 90$ | | | | | |
| FCM-AO | 3 | 0.0245 | 1.2238 | 10.3762 | 1.8856 |
| | 5 | 0.0291 | 2.1925 | 22.4657 | 2.5320 |
| | 10 | 0.0132 | 2.5490 | 43.6115 | 6.9894 |
| CDE-FCM | 3 | 0.0239 | 1.2197 | 9.3250 | 1.7534 |
| **BearingData2**, $n = 90$ | | | | | |
| FCM-AO | 3 | 0.4578 | 17.895 | 71.965 | 5.3372 |
| | 5 | 0.5120 | 23.559 | 123.189 | 7.8900 |
| | 10 | 0.3251 | 14.358 | 53.172 | 4.3256 |
| CDE-FCM | 5 | 0.5052 | 19.856 | 112.885 | 7.2505 |
| **BearingData3**, $n = 90$ | | | | | |
| FCM-AO | 3 | 0.7925 | 8.6690 | 45.890 | 3.897 |
| | 5 | 0.8456 | 12.367 | 119.559 | 4.673 |
| | 10 | 1.6251 | 22.145 | 145.335 | 8.212 |
| CDE-FCM | 3 | 0.8280 | 8.5241 | 36.879 | 3.865 |
| **BearingData4**, $n = 90$ | | | | | |
| FCM-AO | 3 | 0.8955 | 8.356 | 67.189 | 2.9836 |
| | 6 | 0.7341 | 5.3536 | 50.338 | 2.5620 |
| | 10 | 1.3802 | 12.7467 | 121.298 | 3.2461 |
| CDE-FCM | 6 | 0.6821 | 5.1130 | 48.827 | 2.5465 |
| **BearingData5**, $n = 150$ | | | | | |
| FCM-AO | 3 | 0.0026 | 1.2876 | 8.9281 | 0.3653 |
| | 6 | 0.0012 | 1.1926 | 6.7172 | 0.3156 |
| | 10 | 0.0104 | 1.8927 | 11.602 | 0.7253 |
| CDE-FCM | 6 | 0.0011 | 1.2110 | 6.5670 | 0.3267 |

The proposed algorithm CDE-FCM almost always outperforms FCM-AO. In cases where the three natural groupings are not found, CDE-FCM finds approximately 6 clusters which are decoded as two subclusters based on speed (high and low) in most cases. The accuracy, precision, and recall rate of both FCM-AO and CDE-FCM are similar meaning they uncover very similar clusters although with FCM-AO it is often difficult to ascertain the optimal number of clusters.

A comparative evaluation is performed using sensitivity analysis. In clustering evaluation, a true positive (*TP*) is defined as the decision that assigns two similar data objects in

the same cluster and a true negative (*TN*) is a decision that assigns two dissimilar objects to different clusters. These are both desirable outcomes. The errors can either be a false-positive (*FP*) decision for an assignment of two dissimilar objects to the same cluster or a false-negative (*FN*) decision when two similar objects are assigned to different clusters. These metrics are evaluated using pairwise measurements—for a dataset of cardinality $n$, there are $n(n-1)/2$ pairs of objects. The precision (*P*) and recall (*R*) are defined as,

$$P = \frac{TP}{TP + FP}, \ R = \frac{TP}{TP + FN} \tag{26}$$

The Rand index *RI* measures the percentage of decisions that are correct (also called the accuracy). The *F*-score is a measure of the harmonic mean of the method's precision and recall. The *F*-score is implemented as $F_1$ in this paper (which gives equal weight to precision and recall) [67].

$$RI = \frac{TP + TN}{TP + FP + FN + TN}, \ F_1 = \frac{2PR}{P + R} \tag{27}$$

Results of the sensitivity analysis comparing CDE-FCM with FCM-AO are provided in Table 10. As can be seen, the proposed method CDE-FCM achieves accuracies of approximately 75% with two of the five datasets. The best accuracy of FCM-AO approaches 75% for only one of the five datasets. The $F_1$ score equally weighing precision and recall for CDE-FCM is also better than FCM-AO for all five datasets. This is a promising result that shows the superiority of the proposed method over the original FCM with randomized initialization.

**Table 10.** Comparative evaluation of FCM-AO and CDE-FCM for five bearing fault datasets.

| | *TP* | *FP* | *TN* | *FN* | *P* | *R* | *RI* | $F_1$ |
|---|---|---|---|---|---|---|---|---|
| **BearingData1**, $n = 90$, $c = 3$ | | | | | | | | |
| FCM-AO | 1295 | 721 | 1366 | 263 | 0.642 | 0.675 | 0.664 | 0.658 |
| CDE-FCM | 1481 | 694 | 1404 | 426 | 0.681 | 0.777 | 0.720 | 0.725 |
| **BearingData2**, $n = 90$, $c = 5$ | | | | | | | | |
| FCM-AO | 1326 | 1926 | 1228 | 525 | 0.589 | 0.716 | 0.638 | 0.646 |
| CDE-FCM | 1497 | 612 | 1478 | 418 | 0.710 | 0.782 | 0.742 | 0.744 |
| **BearingData3**, $n = 90$, $c = 3$ | | | | | | | | |
| FCM-AO | 1205 | 629 | 1516 | 655 | 0.657 | 0.648 | 0.679 | 0.652 |
| CDE-FCM | 1422 | 598 | 1498 | 487 | 0.704 | 0.745 | 0.729 | 0.724 |
| **BearingData4**, $n = 90$, $c = 6$ | | | | | | | | |
| FCM-AO | 1375 | 826 | 1075 | 729 | 0.625 | 0.654 | 0.612 | 0.639 |
| CDE-FCM | 1457 | 683 | 1190 | 675 | 0.680 | 0.683 | 0.661 | 0.682 |
| **BearingData5**, $n = 150$, $c = 6$ | | | | | | | | |
| FCM-AO | 4219 | 1622 | 4143 | 1191 | 0.772 | 0.780 | 0.748 | 0.750 |
| CDE-FCM | 4522 | 1433 | 4291 | 929 | 0.760 | 0.830 | 0.789 | 0.793 |

## 7. Conclusions and Directions of Future Work

A novel method of initializing cluster prototypes for fuzzy c-means (FCM) is presented in this paper. The method also simultaneously finds the optimal number of clusters in the partition. These two constitute the biggest drawbacks of clustering techniques such as FCM. Many attempts have been made in addressing the two issues-we present a very detailed review of the existing literature in this paper. The concept presented in this paper not only complements the body of work in this field but is also a non-trivial improvement on present

techniques. We propose a co-evolutionary multi-population differential evolution-based technique to evolve a candidate vector that would encode for the most optimal set of initial cluster prototypes to use and also for the optimal number of clusters to find. This is based on a smaller subset of the original data thus reducing effort during evolution. The best subset is chosen based on a sparse sampling statistic from literature. A novel fitness evaluation function based on two cluster validity measures is also proposed. The proposed method called cooperative differential evolution for fuzzy c-means (CDE-FCM) is compared to some state-of-art improvements of FCM and also to the original FCM with random initial cluster prototypes implemented for a range of values for the number of clusters. The comparison is performed using a distinct set of cluster validity indices. In many of the cases with synthetic two-dimensional data and larger dimensional data from the UCI ML repository, the proposed algorithm performs better than the methods compared, and in almost all the cases, is almost as effective as the best of the other methods. The proposed method is also used on a real-world experimental dataset to partition rolling-bearing fault data. The technique worked well in real time and with very minimal improvements can be deployed to a live data analysis project such as this.

For future work, modifications to improve the proposed methodology will be investigated. Although the fitness evaluation and candidate evolution in the multi-population method are not resource intensive for parallel implementation, they can both be improved further. Statistical testing to evaluate the level of significance in performance improvement of the proposed method will also be performed as part of future work. This will result in ranking the comparative clustering methods used in this paper over other datasets from the UCI ML repository and on other benchmark datasets.

**Author Contributions:** Conceptualization, A.B.; methodology, A.B.; software, A.B. and I.A.-M.; validation, A.B.; formal analysis, A.B.; investigation, A.B. and I.A.-M.; resources, A.B. and I.A.-M.; data curation, A.B. and I.A.-M.; writing—original draft preparation, A.B.; writing—review and editing, A.B. and I.A.-M.; visualization, A.B. and I.A.-M.; supervision, A.B.; project administration, A.B. and I.A.-M. All authors have read and agreed to the published version of the manuscript.

**Funding:** This research received no external funding.

**Data Availability Statement:** Data used in this study can be made available by request.

**Conflicts of Interest:** The authors declare no conflict of interest.

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
