# Peer review of "A Novel Adaptive FCM with Cooperative Multi-Population Differential Evolution Optimization"

_algorithms, doi:10.3390/a15100380_

Round 1

Reviewer 1 Report

In this paper, the authors introduce A Novel Adaptive FCM with

Cooperative Multi-Population Differential Evolution Optimization.

Comments:

1.     Contributions of this paper need to summarize and each contribution point start from new line.

2.     The paper is too wordy. The readability of the paper should be significantly improved.

3.     In Background and others section, authors mentioned some equations but did not mention reference for these equations. Need to mention references.

4.     In Simulation and Results section, my observation is that if authors use graphical representation for showing Performance of Algorithms, it will be better for other researchers to understand the paper contributions.   

Author Response

Thank you very much for your review. Below is our response

  1. Contributions of this paper need to summarize and each contribution point start from new line.

We have listed three main contributions very clearly in section 1. Each contribution point starts from a new line as suggested.  

  1. The paper is too wordy. The readability of the paper should be significantly improved.

We have tried to be succinct, but the information presented needs to be clearly explained. We feel that although it may look wordy, it is not superfluous.

  1. In Background and others section, authors mentioned some equations but did not mention reference for these equations. Need to mention references.

We have added references to all the equations used in the paper. Thank you for pointing it out.  

  1. In Simulation and Results section, my observation is that if authors use graphical representation for showing Performance of Algorithms, it will be better for other researchers to understand the paper contributions.   

We can show results of convergence performance of various algorithms graphically, but the main distinction is the cluster validity indices for different values of number of clusters c and this can be presented as tabular information as done in Tables 3-7.

Reviewer 2 Report

In this paper, the authors proposed a new cooperative multi- 12 population differential evolution method with elitism to identify near optimal initial cluster 13 prototypes and also determine the most optimal number of clusters in the data. The method overcome two existed specific drawbacks. And some simulations are given to show the efficiency of their method. The paper can be published in Algorithms.

Some minor suggestions:

Please revise some references, for example 13(no page number), 18(delete Jun. 2015. ) and so on.

Author Response

Some minor suggestions: Please revise some references, for example 13(no page number), 18(delete Jun. 2015) and so on.

Thank you very much for your review. We have revised the references and added/deleted information as needed.

Reviewer 3 Report

This paper proposes a new cooperative multi-population differential evolution method with elitism to identify near optimal initial cluster prototypes and also determine the most optimal number of clusters in the data.

Although the paper is mostly organized well, some parts must be revised before making final decision. 

First, in Steps 3 and 4 of Algorithm 1, `SP' must be `(n-rem)/SP'.

Second, in Step 10 of Algorithm 2, the stopping condition is not fair. 

Third, in the last of Section 6, the authors said `A more detailed analysis will be included in the final version of this paper'. 

However, all contents should be reviewed before making final decision. 

So, the additional part must be presented and the paper should be re-reviewed. 

Author Response

Thank you for the detailed review. We are really appreciative. Below are our responses to your review comments. 

  1. First, in Steps 3 and 4 of Algorithm 1, `SP' must be `(n-rem)/SP'.

Thank you for catching the error. We have now corrected it.

  1. Second, in Step 10 of Algorithm 2, the stopping condition is not fair. 

The process of selecting the best subset is repeated until the difference between the Hopkins statistic value for the entire dataset and any subset is less than the threshold ε. The algorithm stops when the subset that has the smallest difference is identified.

  1. Third, in the last of Section 6, the authors said `A more detailed analysis will be included in the final version of this paper'. However, all contents should be reviewed before making final decision. 

We have included a sensitivity (ROC) analysis for clustering evaluation in section 6. This is a little more complex for an n-class clustering if n ≠ 2. This is clearly presented at the end of section 6. The proposed method achieves better true positive and true negative rates than FCM-AO. This is presented in Table 9.  

Round 2

Reviewer 3 Report

The reviewer believes that the paper is now acceptable as it is.